# An Investigation into Current Sand Control Methodologies Taking into Account Geomechanical, Field and Laboratory Data Analysis

**Dmitry Tananykhin** [1,*] , **Maxim Korolev** [2] , **Ilya Stecyuk** [2] **and Maxim Grigorev** [1]

1 Petroleum Faculty, Saint Petersburg Mining University, 199106 Saint Petersburg, Russia; makcum1298@mail.ru
2 Oil and Gas Production Department, LLC RN-Purneftegaz, 629830 Gubkinskiy, Russia; korolevhik@yandex.ru (M.K.); iastetsyuk@png.rosneft.ru (I.S.)
* Correspondence: dmitryspmi@mail.ru

**Abstract:** Sand production is one of the major issues in the development of reservoirs in poorly cemented rocks. Geomechanical modeling gives us an opportunity to calculate the reservoir stress state, a major parameter that determines the stable pressure required in the bottomhole formation zone to prevent sand production, decrease the likelihood of a well collapse and address other important challenges. Field data regarding the influence of water cut, bottomhole pressure and fluid flow rate on the amount of sand produced was compiled and analyzed. Geomechanical stress-state models and Llade's criterion were constructed and applied to confirm the high likelihood of sanding in future wells using the Mohr–Coulomb and Mogi–Coulomb prototypes. In many applications, the destruction of the bottomhole zone cannot be solved using well mode operations. In such cases, it is necessary to perform sand retention or prepack tests in order to choose the most appropriate technology. The authors of this paper conducted a series of laboratory prepack tests and it was found that sanding is quite a dynamic process and that the most significant sand production occurs in the early stages of well operation. With time, the amount of produced sand decreases greatly—up to 20 times following the production of 6 pore volumes. Finally, the authors formulated a methodological approach to sand-free oil production.

**Keywords:** sanding; sand control; poorly consolidated reservoir; prepack test; slotted liner; geomechanical modeling

## 1. Introduction

The process of sand production is often associated with the development of poorly cemented reservoirs. The first reservoir equilibrium stress state is already reached during the drilling process, and becomes more severe with further well operation. As a result, the destruction of rocks in the bottomhole zone occurs when stresses exceed their tensile strengths [1–3]. This leads to an increased concentration of suspended rock particles in produced liquid, causing submersible and surface equipment malfunctions. Resulting in a decreased well operation factor due to an increase in the frequency and duration of repairs and, as a result, in operating costs [4–7].

There are three main initiation mechanisms of rock destruction. Two of them consist of violating rock integrity by exceeding their compressive or tensile strengths with shear and tensile stresses, respectively. The dynamics of sand production as a result of tensile stresses is, as a rule, short-term, rapidly decaying and local in character and does not lead to significant difficulties during well operation. The third mechanism is associated with volumetric destruction of pore space and is currently poorly studied due to the complexity of the physical processes and the difficulties associated with clear formalization of the task due to multiple influencing factors [1].

As a result of this process, a plastic zone that grows with time is formed around the perforations, associated with the appearance of permanent deformations, the mechanical and reservoir properties of which differ from the remote part of the formation.

Calculation methods based on geomechanical data should be used to prevent the occurrence of critical stresses, at which the destruction process of the bottomhole zone intensifies. One of these methods is selecting the optimal drawdown for sand free operation of the well. The prediction of critical drawdown is formalized by a problem solved using geomechanical modeling [8–15]. Modeling the stability of the bottomhole formation zone (BHZ) makes it possible to predict potential complications during well operation associated with the mechanical properties of the rocks. Such models are used to determine the optimal well completion and magnitude of the sand-free drawdown along with the location and orientation of perforations [16–18].

In order to build the model, the following data are necessary: well logging data, the results of core studies (compressive strength, static and dynamic Young's moduli, Poisson's ratio), as well as operating and drilling data.

The purpose of this research is to increase the turnaround time of the well due to the development of an algorithm for selecting the optimal operating parameters of the well, in conditions of sand removal from the formation by taking into account geomechanical, field and laboratory data analysis.

*Technologies for the Operation of Wells Complicated by Sand Production*

There are two general technological approaches used in combating the sand production process:

(a)   Preventing the ingress of mechanical inclusions;
(b)   Allowing and working with the consequences rock particle ingress into the wellbore [19,20].

Both approaches are actively being developed, which is confirmed by numerous publications on the use of various technologies in many Russian (and other) fields: Russkoye, Messoyakhskaya group of fields, Van-Yeganskoye, Medvezhye, Komsomolskoye, Vankorskoye, etc. The photo of the region (depositional environment) where the oilfield discussed in this article is located can be seen in Figure 1.

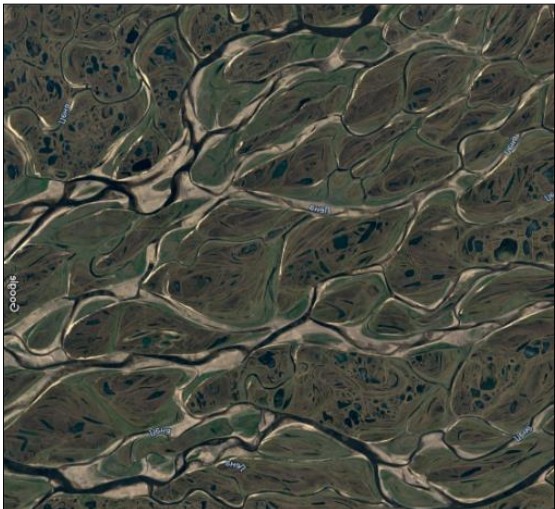

**Figure 1.** Satellite photo of the area.

Reservoir deposits are identified within the upper part of the Pokurskaya site and are characterized by tidal Upper Cretaceous sediments of the Cenomanian stage, represented by weakly compacted rocks: sands, sandstones, silts, siltstones and argillites (mudstones). The deposits are characterized by explicit facies heterogeneity. The above-mentioned

technologies show variable efficiency, and their application is accompanied by significant disadvantages, for example:

- The use of screens causes stress destabilization in the bottomhole zone;
- There is an increase in the extra skin factor ranging from 2 to 10;
- There is a need for their periodic replacement/cleaning (due to erosion wear);
- The use of chemical compositions for fixing the bottomhole zone can reduce the permeability (in some cases up to 70%) due to clogging of highly permeable channels (since the injected composition enters them first), and they also operate for a limited period of time;
- Gravel packing is not always possible (for example, in horizontal wells), and where used, imposes restrictions on the completion of the well;
- Specific gravel pack assemblies require either carefully graded gravel or specially prepared gravel (which is more expensive in terms of its applicability in horizontal wells).

There is also an operational method for limiting sand production—regulating the technological parameters of the well operation, which consists of reducing the depression to the minimum permissible values in order to prevent the ingress of rock particles into the well, but its disadvantage is quite clear—an artificially low flow rate [21–24].

In the case of high-viscosity oil, these factors are exacerbated many times over due to the low productivity index (PI) of the well, which has made the sand management approach the subject of some interest [25–27].

This approach consists of two aspects: careful and constant monitoring of the operating parameters of individual wells and optimization of risks (in the form of predictive calculations and modeling) that inevitably appear when rock is removed from the formation without control.

Given these aspects, in developing this approach, it was understood that thorough consideration of each well was necessary in order to obtain a general situational understanding [28–32].

Additionally, some predictive analytics methods were studied, consisting of calculations regarding:

- Predicting the initialization time of the sand development process;
- The volume of sand production;
- The ability of the rock particles to migrate in the bottomhole zone.

The above-mentioned technique requires the analysis of a vast amount of data, including both the formation properties and well parameters.

Therefore, we formulated an approach based on the studying the influencing parameters on the sand production process: pH of formation water, oil and liquid production rate, water cut, number of shutdowns and starts of wells, aperture of installed downhole screen, method of well completion, reservoir and bottomhole pressure, drawdown, well arrangement, particle size distribution and many others.

## 2. Materials and Methods

### 2.1. Geomechanical Modeling

Many researchers [33–43] find a notably high influence of fine fractions (<50 μm) on the well operation (mainly plugging screens), as a result, the material associated with the formation of sand arches was worked out, but significant results were not achieved in by analyzing the literature (except for the connection of the aforementioned arches with the process of the natural decrease in the number of suspended particles in the first few days of well operation).

Some investigations look at the use of chemical compositions for the selective retention of fine fractions; however, no field test results have been conducted [14,15,44–48].

Interdependences were investigated within the framework of a field with high viscosity oil, currently under development between the following parameters: the number of suspended particles, the influence of the sand production process on the operation of

individual wells and the study of the effectiveness of screens of certain standard sizes and structures [49,50]. It was found that with different production rates and water-cuts, the interdependence of the number of suspended particles on certain parameters may or may not be observed, as, for example, with the flow rate: its increase or decrease does not affect the number of suspended particles in the fluid flow, which is unexpected (since the fluid flow with a higher speed should entrain more rock particles from the formation). It was found that for wells with a liquid flow rate of <100 $m^3$/day, there is a significant dependence for the number of suspended particles on water cut (for wells with a flow rate >100 $m^3$/day, there is no dependence). The effect of water cut on the process of formation destruction has been noted numerous times, which is some confirmation of the fact that phase flow is one of the key parameters that should be taken into account when working with a poorly cemented reservoir.

The authors analyzed the operating experience of one of the facilities and constructed a distribution of the well stock, categorizing wells as either "complicated" or "uncomplicated" according to the following criteria: flow rates (Figure 2), water cut (Figure 3) and target bottomhole pressure (Figure 4). It can be seen from the graphs that operational complications (failures due to erosive wear or clogging with mechanical inclusions of downhole pumping equipment are mostly observed in the well stock with a flow rate of less than 100 $m^3$/day and a water cut of less than 50%).

Coupling of wells is not always applicable due to the differentials in the tubing diameter, equipment and other factors, which influence the sanding process. The number of wells for consideration for the first category (0–50 $m^3$/day) is three times fewer than the second. Making conclusions based on the beforementioned data seems questionable since a two-time increase in liquid flow rate leads to a rise in sanding. Nevertheless, a further increase in flow rate does not lead to complications in the well.

It is worth noting that many authors have found during their investigations that the amount of sand carried out increases along with an increase in water cut. Nevertheless, the graph above does not show this effect, since when the water cut is above 50%, there are no complicated wells at all.

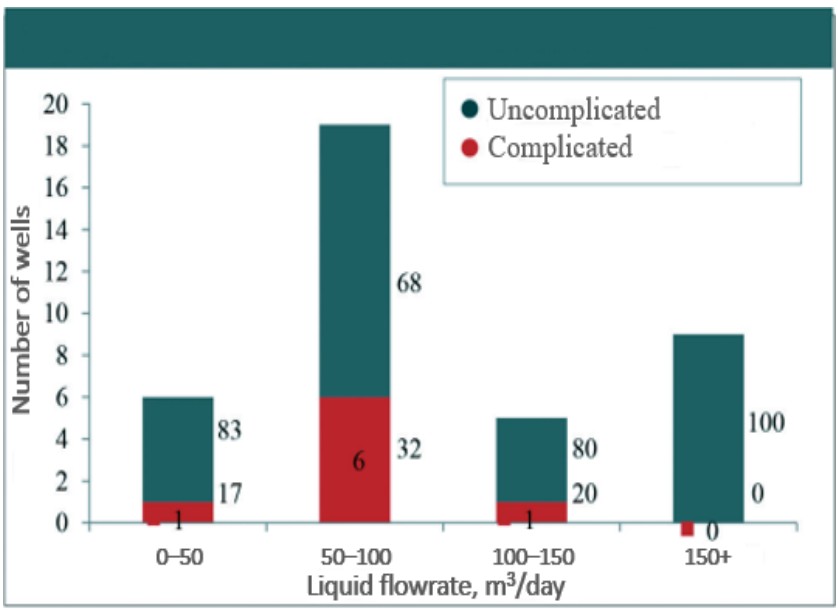

**Figure 2.** Distribution of sand-prone well stock by fluid flow rate.

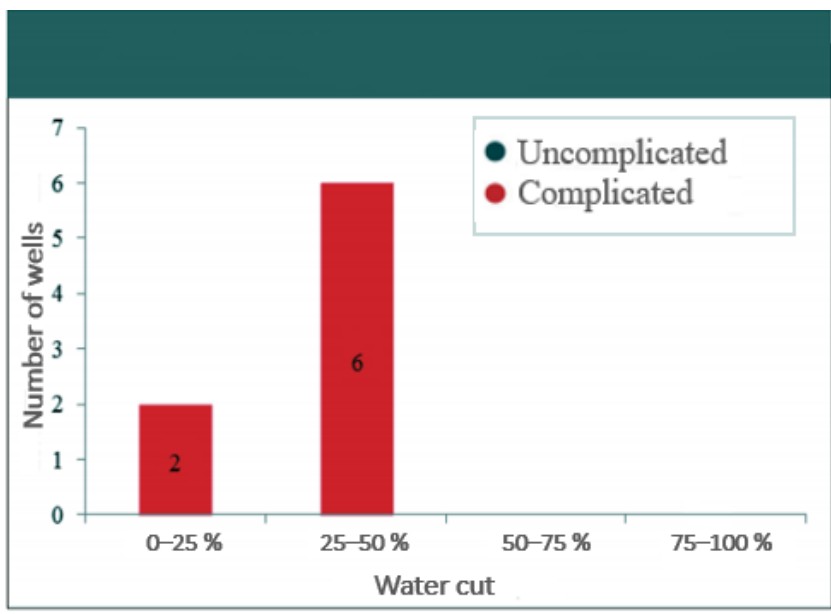

**Figure 3.** Distribution of sand-prone well stock by water cut.

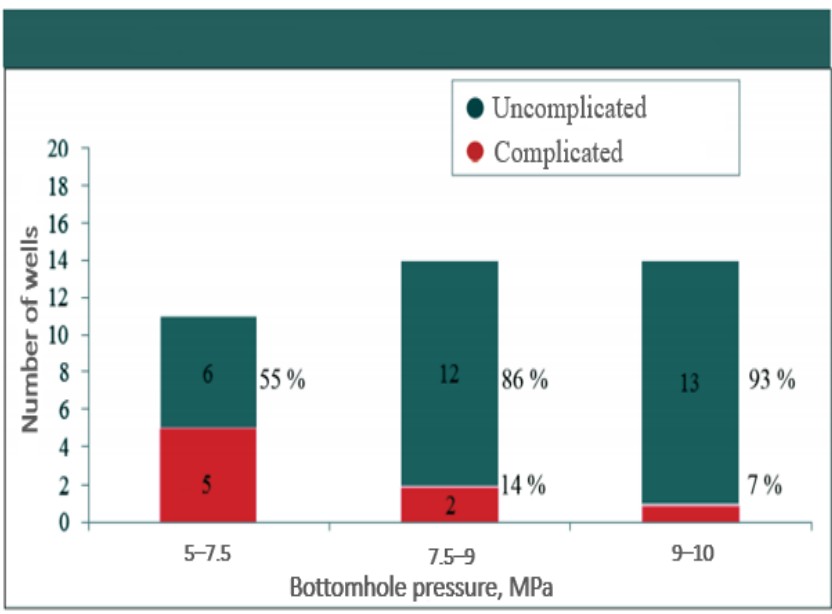

**Figure 4.** Distribution of sand-prone well stock by bottomhole pressure.

With a decrease in bottomhole pressure (on average, a higher depression of the reservoir), a dependence of a higher level of sanding and complication can be observed.

This phenomenon is associated with the low bearing capacity of the flow and with high viscosity of the oil, which is explained by the lower sedimentation rate of the sand particles. Well operation is carried out according to the target bottomhole pressure control program, which is determined by the requirements of rational oilfield.

The trend towards lower drawdowns in the bottomhole formation zone is confirmed by the analysis of Figure 4, where it can be seen that the least number of failures and complications in the well stock with a target bottomhole pressure of more than 9 MPa, with the initial formation pressure—10.6 MPa.

Thus, the main issue of scientific and practical interest is the prediction of the onset of reservoir destruction and further determination of the critical bottomhole pressure (which leads to the production of sand together with the formation fluid) and, ultimately, to finding

the optimal dynamics of bottomhole pressure lowering. There are two basic ways to solve this issue—practical modeling (laboratory tests) via sand retention tests (SRT) and Prepack tests and mathematical modeling. Physical modeling gives good results and provides a lot of information; however, preparing these experiments is time consuming, especially with bulk modeling included.

A literature review [16–19] makes it possible to recommend geomechanical modeling as a tool for assessing the stability of the bottomhole formation zone during operation in the conditions of weakly consolidated sandstones. The stability model is adapted to the data of caliper, imager, mini-frac and modular dynamic tests (MDT) studies. The model-building sequence for one-dimensional geomechanical modeling for PK formations is shown in Figure 5.

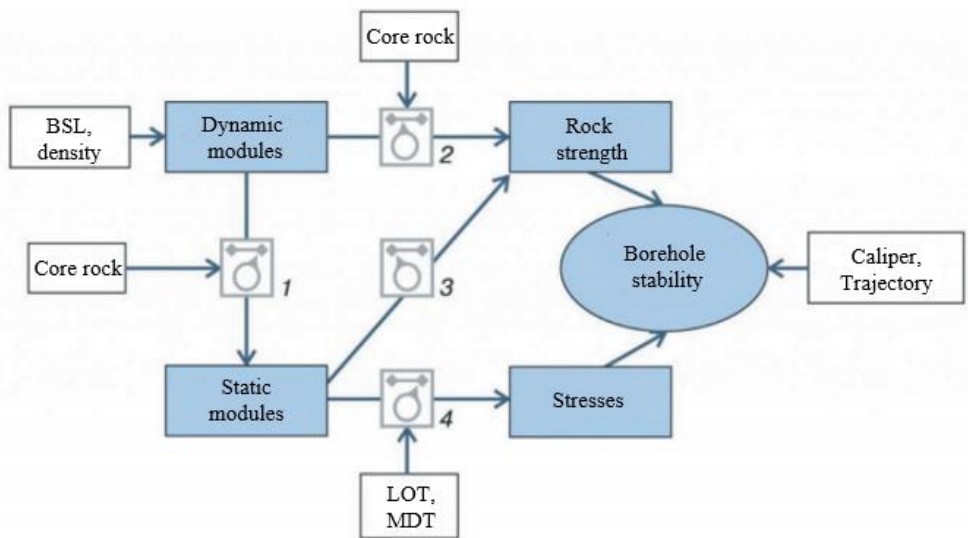

**Figure 5.** General scheme of geomechanical modeling. BSL—Broadband sonic logging, LOT—fractest, MDT—stress-test with bottomhole tester.

The 1D model of stability of the bottomhole formation zone according to the criteria of Mogi–Coulomb and Mohr–Coulomb is based on the current parameters of the formation and according to the data of the well operation:

- Reservoir pressure;
- Vertical stress;
- Minimum and maximum horizontal stresses;
- Adhesion strength of the rock;
- Angle of friction;
- Borehole azimuth;
- Well profile;
- Biot's poroelastic constant;
- Poisson's ratio.

The bottomhole pressure and the angular position around the circumference of the wellbore are the specified parameters in this model. Detailed algorithm for this modeling procedure is presented in [9].

The results of the calculations of the 1D model are used to determine the admissible value of depression, at which the fracture of the bottomhole formation zone will not occur. The next stage is a calculation of the optimal step for lowering the bottomhole pressure to value, when the well is brought to the target operating mode (at analogous fields— 0.3–0.5 MPa/day). A significant advantage of the proposed approach is the ability to assess the critical depression value even in the absence of core studies of the mechanical properties of the rock.

As a result, the following geomechanical modeling algorithm was developed:

1.  Construction of the one-dimensional model of mechanical properties using well logging and standard correlations;
2.  Calculation of stresses and adaptation of the minimum stress to the data on mini-frac;
3.  Adaptation of the maximum horizontal stress and strength along the profile of the caliper;
4.  Calculation of the critical depression profile based on the correlations set in the software and adaptation of strength based on the development and operation history of previously perforated intervals;
5.  Forecast of critical depressions for intervals.

### 2.2. Prepack Test Design

The methodology for the current prepack tests series was developed to simulate reservoir conditions in the lab with different screens, flowrate conditions and drawdowns. The same differentials as in Figures 1–3 were chosen to be variables in the tests. Thus, tests were run with different water/gas cuts (30, 50, 90%), different drawdowns (gradP1 and gradP2, which were four times higher than gradP1).

Slotted liner was chosen to be tested in this series due to its simplicity, availability on the market and prevalence among Russian oil and gas companies. Screens with aperture sizes of 100, 150, 200, 500, 700 and 1000 μm (mcm) were tested. Initial test runs showed that the 1000 mcm screen was inappropriate for the testing conditions. A schematic representation of the testing facility is shown in Figure 6. The coreholder is equipped with a cuff, and the crimp pressure was set to 2.04 MPa.

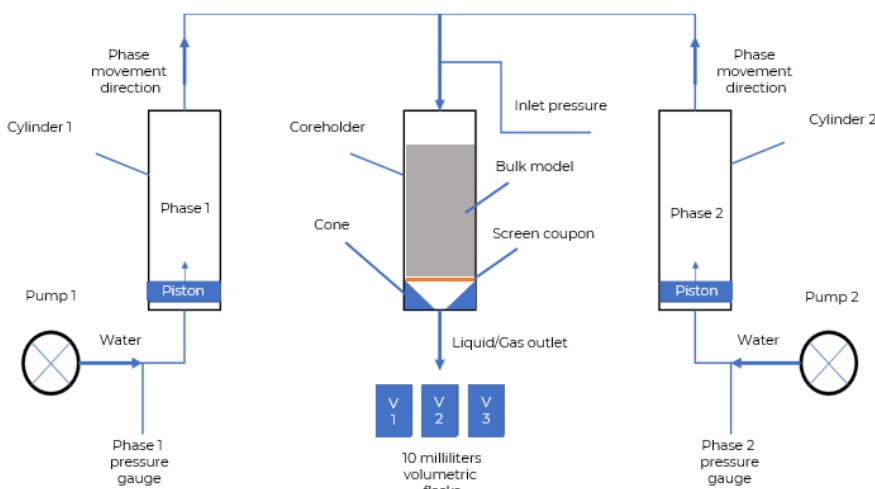

**Figure 6.** Prepack testing facility.

Bulk models were made with intention to reach porosity, permeability and particle size distribution (PSD) mirroring that of reservoirs. The original PSD curve was taken from one of the oilfields and corresponded to the PK1 formation, shown in Figure 7. Sand from oilfield was used as a material to make bulk models. It was pre-extracted with solvent flushing (until solvent was transparent), dried at 60 degrees Celsius and then sorted using sieves (section sizes of 100, 125, 160, 215, 250, 315 and 500 μm).

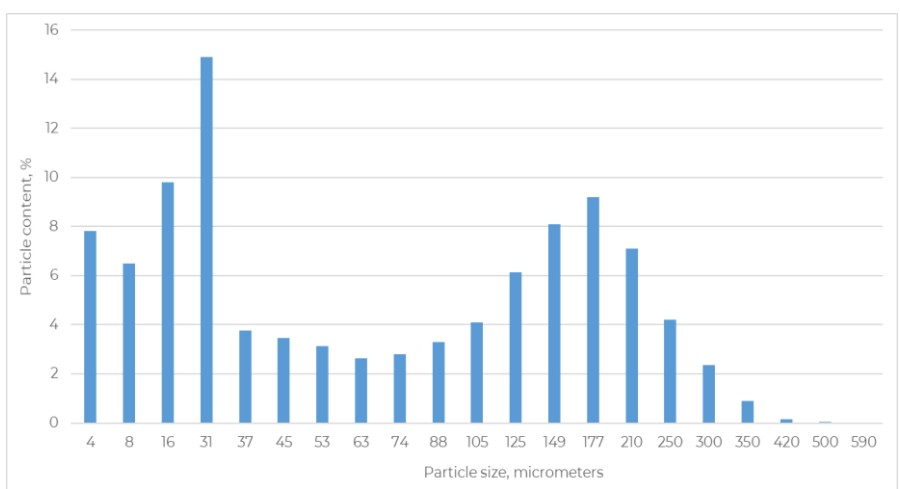

**Figure 7.** Reservoir's PSD curve.

Bulk models (Figure 8) were made with moist tamping technique, while brine was used as liquid to dampen the models. The diameter of the model is 3 cm, height varied from 5 to 7 cm. Vacuum treatment in a jar with mineral oil was then applied to reach the initial saturation parameters for brine and oil. Mineral oil of the appropriate viscosity (75 mPa*s) was later used as a model of oil in the experiments.

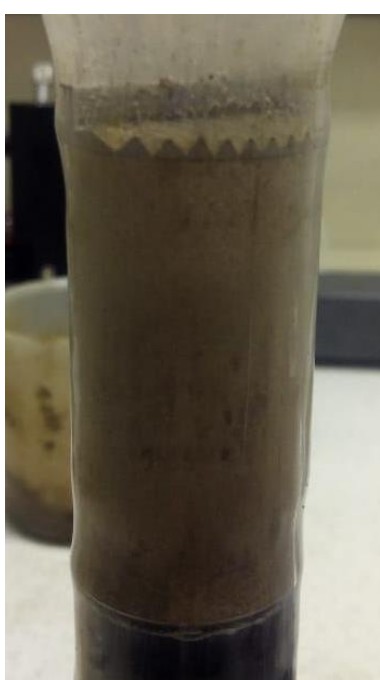

**Figure 8.** Bulk model with screen filter disk installed.

Samples of liquid were taken three times in a row using volumetric flasks (10 mL) to get dynamic data on the number of suspended particles. Suspended solids concentration (SPC) was then calculated with the mass method. Later, the sand was extracted and analyzed for PSD with a laser in-line particle analyzer.

## 3. Results

### 3.1. Geomechanical Modeling

Large reservoirs under the development of LLC "RN-Purneftegaz" were selected as test subjects for geomechanical modeling; namely, the PK1 oil and gas-condensate field

reservoir, as well as reservoirs PK18 and PK19-20 of the oil- and gas-condensate field. Here, PK is the name of the formation, and the numbers refer to the number of a single layer in the entire formation.

The developed reservoirs are uncemented sandstone, so the equipment operates under conditions of increased abrasive wear. The removal of mechanical inclusions (mainly sand) is very significant from 3 mg/L to 2050 mg/L (average 109 mg/L) for the PK1 reservoir and 5.5–1080 mg/L (average 81 mg/L) for the PK19-20. Table 1 shows the geomechanical properties of the reservoirs under consideration within the Pokurskaya site, used for the calculation.

**Table 1.** Reservoir parameters for oil fields.

| Parameter | Oilfield 1 | Oilfield 2 | Oilfield 3 |
|---|---|---|---|
| Reservoir pressure, MPa | 10.3 | 12.0 | 10.8 |
| Rock strength, MPa | 5.6 | 5.6 | 6.5 |
| Vertical stress, MPa | 23.0 | 22.9 | 21.1 |
| Maximum horizontal stress, MPa | 17.7 | 21.5 | 17.5 |
| Minimum horizontal stress, MPa | 16.0 | 19.6 | 16.4 |
| Rock cohesion strength, MPa | 0.22 | 0.29 | 0.24 |
| Friction angle, deg | 24 | 27 | 32 |
| Well's azimuth, deg | 210 | 210 | 329 |
| Well deviation from vertical axis (zenith angle), deg | 89.9 | 89.5 | 89.7 |
| Biot's constant | 0.8 | 1 | 1 |
| Poisson's ratio | 0.31 | 0.2 | 0.32 |

The simulation results are presented in Figures 5–7, where it can be seen that, according to the Mogi–Coulomb and Mohr–Coulomb criteria, the fracture of the bottomhole formation zone will occur even with a minimum pressure drop of 0.1 MPa for all the reservoirs studied (blue line—rock strength, red—current stress). This is also confirmed by the Leid criterion, if $\Delta\sigma_1$ and $\Delta\sigma_3 > 0$, rock destruction should be expected (Table 2). These results indicate that it is imperative to substantiate the technology to prevent the destruction of rocks in the bottomhole formation zone or to deal with sand production in the well.

**Table 2.** Additional parameters.

| Parameter | Oilfield 1 | Oilfield 2 | Oilfield 3 |
|---|---|---|---|
| $\Delta\sigma_1$ | 91.1 | 89.6 | 81.7 |
| $\Delta\sigma_3$ | 84.4 | 200.6 | −1.85 |

The previous statements are also confirmed by Lade's criterion (Table 2).

According to Lade's research and modeling, if $\Delta\sigma_1$ and $\Delta\sigma_3 > 0$, rock destruction will occur. In Table 2, $\sigma_1$ is the significant principal effective stress and $\sigma_3$ is the minor principal effective stress.

These results indicate that it is imperative to substantiate the technology to prevent the destruction of rocks in the bottomhole formation zone or deal with sand production in the well.

The simulation results are presented in Figures 9–11, where we can observe that, according to the Mogi–Coulomb and Mohr–Coulomb criteria, the fracture of the bottomhole formation zone will occur even with a minimum pressure drop of 0.1 MPa for all the presented reservoirs (blue line—rock's strength, red—current stress).

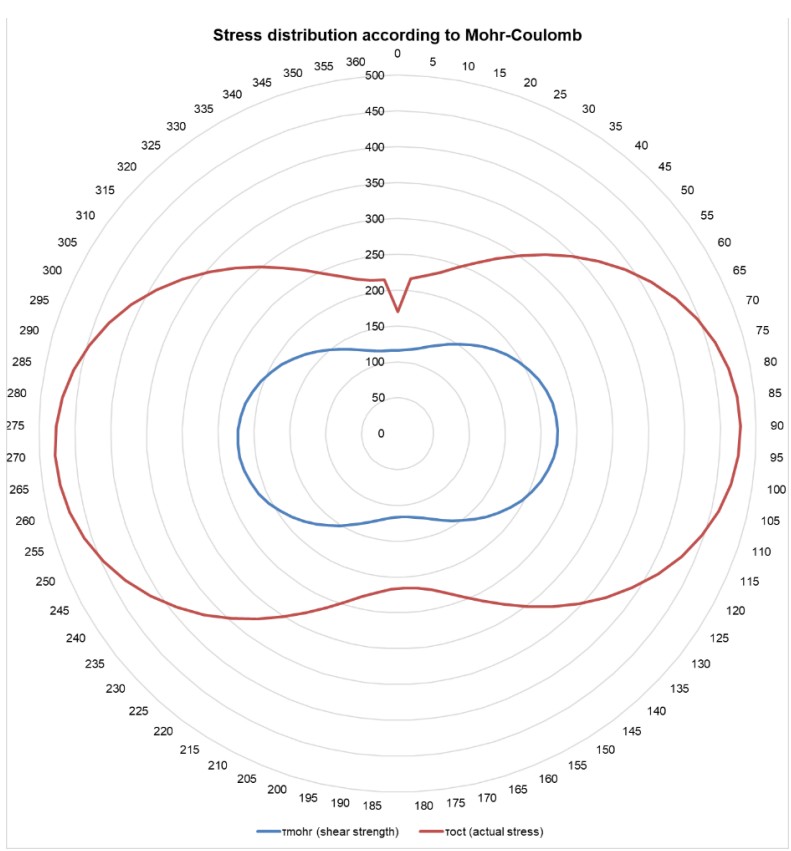

**Figure 9.** Calculated stresses for Field 1 (blue line—shear strength, red line—actual stress).

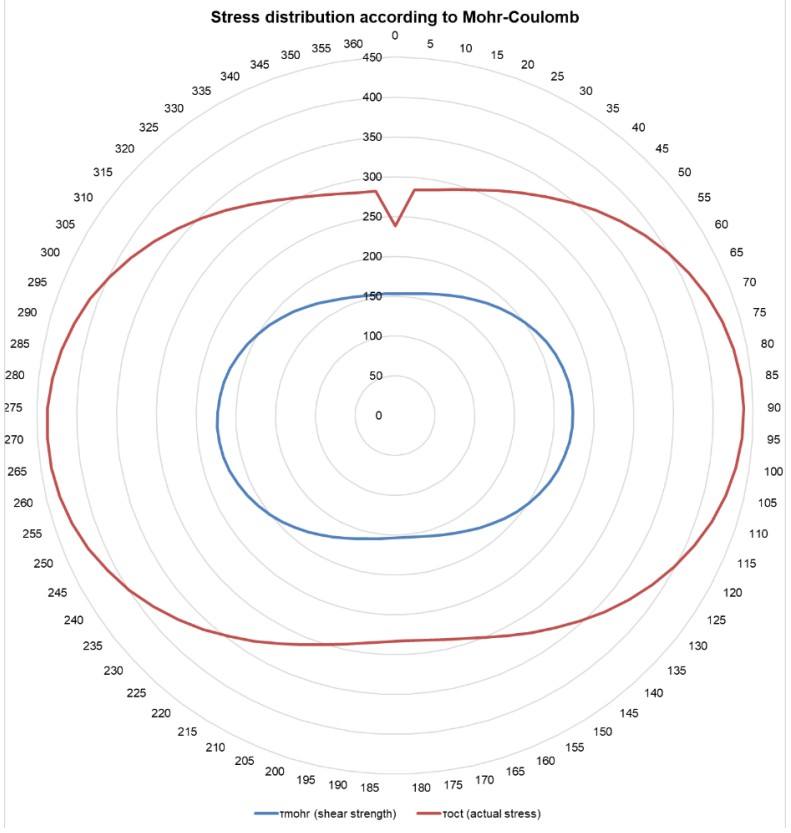

**Figure 10.** Calculated stresses for Field 2 (blue line—shear strength, red line—actual stress).

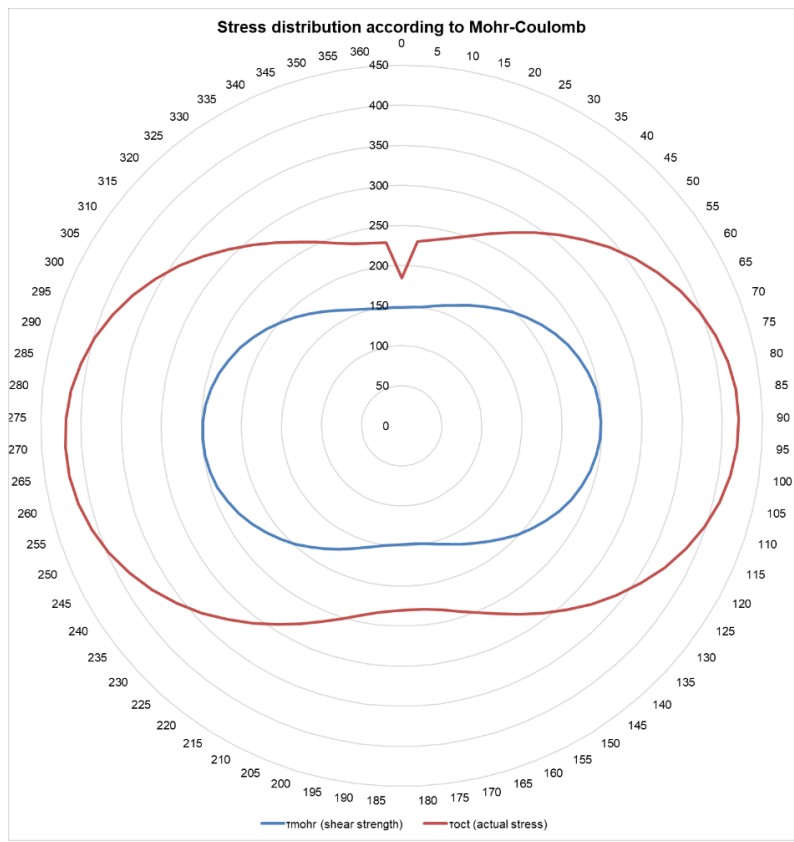

**Figure 11.** Calculated stresses for Field 3 (blue line—shear strength, red line—actual stress).

Sand control with technological restrictions cannot be applied with such reservoir conditions. In this case, it is crucial to ensure that a proper screen or other sand control method will be suitable and will work with maximum efficiency. An inappropriate screen can lead to severe sand influx into the well or decrease permeability in the bottomhole.

*3.2. Prepack Tests*

Since samples were taken sequentially three times at the beginning of the experiment, we had the opportunity to track the change in the number of carried particles for the first 30 mL of the pumped fluid. In almost every experiment, SSC was smaller in the latter stages than during stage 1. This indicates and confirms theories that the most severe sand influx happens during well stimulation and later the amount of sand decreases dramatically. The remaining fluid was collected in a separate container and was not analyzed further, but visual observation showed that the amount of suspended particles in it was minimal, being almost transparent.

Overall, the amount of pumped liquid in Oil/Water experiments was always 250 mL and changed from 25 to 175 mL in experiments with gas. An example of the data obtained is shown in Figure 12 below.

For example, for experiment "O/W 30/70 gradP1" SPC at stage 2, there was only 40% of SPC in first stage, which later decreased to 13% of SPC at stage 1.

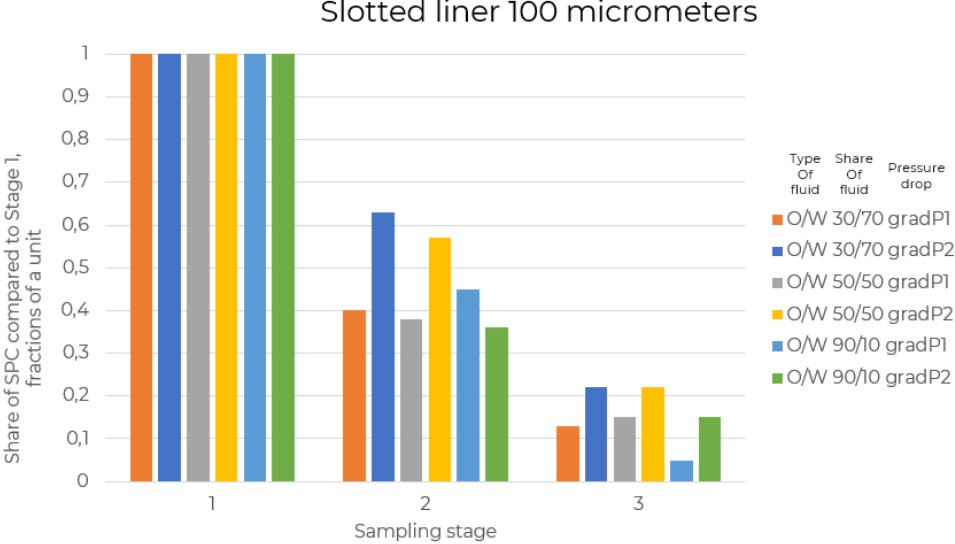

**Figure 12.** Results of the experiments with oil and water in different water cuts (shares are respective).

## 4. Discussion

Theoretical and laboratory studies had been carried out to identify the reasons for the removal of mechanical inclusions from the bottomhole formation zone, as well as methods to prevent the destruction of the reservoir formation. The findings of this research make it possible to develop a methodology for an algorithm of geomechanical modeling and subsequently to recommend the optimal parameters for bringing a well into operation.

The results of 1D-geomechanical modeling confirm the hypothesis about the destruction of the bottomhole formation zone at the objects of the Pokurskaya site both during drilling and maintenance and workover of wells and during well operation. The analysis of field well's failures showed that 84% of mechanical impurities or erosion failures occurred on the first or second voyage of equipment during the process of bringing the well into operation with lower bottomhole pressure at 0.5–1 MPa/day. Furthermore, data also suggest that 61.5% of failures occurred after well shutdowns (by production limitation, workovers, etc.). Therefore, it is necessary to take into account the geomechanical properties of the rock when planning the development of such reservoirs.

In cases where it is not possible to solve the problem of sand influx by technical means, sand control technologies must be employed. The best way to thoroughly investigate the efficiency of the proposed technology is testing in lab conditions by simulation reservoir parameters.

## 5. Conclusions

Using the methodology proposed in the article, the authors noticed a significant decrease in the amount of sand produced after filtration of 3–6 pore volumes. As a result of the implementation of a complex methodology, which included literature analysis, field data analysis, geomechanical modeling and lab testing, the authors developed the following recommendations:

1.  Carrying out 1D and 3D geomechanical modeling in order to clarify the drilling parameters, i.e., permissible bottomhole pressure over reservoir pressure on the formation and the rock penetration rate during the drilling;
2.  The well completion method should be selected from the operating experience of similar objects, using the endings in assemblies with downhole screens;
3.  Bringing the well into operation should be carried out with a minimum gradient of lowering bottomhole pressure—0.2–0.5 MPa/day—which is confirmed by the experience of similar objects' operation.

4. During the operation of the well, considering the possibility of using technologies designed to prevent the removal of mechanical inclusions from the formation.

## 6. Patents

One of the basic elements of this work is a computer program written by the authors "A program for calculating stability criteria and rupture pressures during the operation of wells complicated by sand occurrence" (RU 2020611693).

**Author Contributions:** D.T.—General expertise and lab tests designing; M.K.—Field data gathering, analysis and modeling; I.S.—Field data analysis; M.G.—Modeling, lab tests designing and performing of the experiments. All authors have read and agreed to the published version of the manuscript.

**Funding:** The research was performed at the expense of the subsidy for the state assignment in the field of scientific activity for 2021 № FSRW-2020-0014.

**Institutional Review Board Statement:** Not applicable.

**Informed Consent Statement:** Not applicable.

**Data Availability Statement:** All data included in the article are available for public usage.

**Acknowledgments:** We would like to acknowledge the Saint Petersburg Mining University for the opportunity to work with their equipment and perform experiments and allowing us to use the Research Equipment Sharing Center of the Mining University for probe analysis for the PSD curves.

**Conflicts of Interest:** The authors declare no conflict of interest.

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
