# Peer review of "An Investigation into Current Sand Control Methodologies Taking into Account Geomechanical, Field and Laboratory Data Analysis"

_resources, doi:10.3390/resources10120125_

Round 1
Reviewer 1 Report
The paper ‘An investigation into current sand control methodologies taking into account geomechanical, field and laboratory data analysis’ is very interesting. It shows the results of research in the important field of petroleum engineering. Moderate editing of English language and style required.
Some improvement is necessary, and the things that should be revised are as follows.
- Line 17 – Llade (mistake in name) criterion – This sentence is not discussed in the main text.
- Not all abbreviations are explained in its first appearance.
- Please explain the meaning of physical symbols.
- Please check the formatting carefully – some subscripts are not presented correctly, using small letter for name in the figures, repeating figures number.
- Each figure and table should be mentioned in the main text before first appearance and discussed later on.
- It is not clear for the reader, is Pokurskaya the only considered field? What is Field 1, 2, and 3? Please make it clear in the main text.
- ‘Stress-state model’ is much more than presented in the Figs. 5-7. Please correct titles and explain in the main text what is presented in each chart.
- Conclusion 1. – What do you mean by “permissible repressions”? – There is no sufficient discussion in the main text.
- Conclusions 3 and 4. – There is no sufficient discussion in the main text.
Author Response
To whom it may concern,
We hope you and your family are healthy and safe during these uncertain and unprecedented times.
You can see all the changes that were made in the editor’s mode of MS Office Word.
Since there is more than one editor, you can see more changes than each single editor suggested or proposed in his/her review.
We will once again check the paper for mistakes and format issues;
We have asked one of our colleagues with high level of English to check the paper for mistakes. His corrections have been taken into account.
Response to Reviewer 1 Comments
Point 1: Line 17 – Llade (mistake in name) criterion – This sentence is not discussed in the main text.
Response 1: The mistake has been corrected. Even though the criterion itself is not discussed, there are calculations based on that criterion (Table 2) and once again the Lade’s criterion is mentioned in lines 259-261.
Point 2: Not all abbreviations are explained in its first appearance.
Response 2: Abbreviations have been added. Suspended particles coefficient (translation of Russian abbreviation) was changed to Suspended Solids Content (SSC). Which we think is a better fit.
Point 3: Please explain the meaning of physical symbols.
Response 3: Explanations for the symbols σ1 and σ3 are added.
Point 4: Please check the formatting carefully – some subscripts are not presented correctly, using small letter for name in the figures, repeating figures number.
Response 4: The numbers have been corrected, but we couldn’t find formatting issues with using small letters in the names.
Point 5: Each figure and table should be mentioned in the main text before first appearance and discussed later on.
Response 5: We added comments for Figures 2-4.
Figures 5-11 are self-explanatory and are mentioned in the text before they appear. They either visualize data or give some visual explanations of the text above them.
Point 6: It is not clear for the reader, is Pokurskaya the only considered field? What is Field 1, 2, and 3? Please make it clear in the main text.
Response 6: We should note that Pokurskaya is not the oilfield, but one of the formations in the stratigraphy of the region. We should also clarify that objects PK1, PK19-20 can be explained in the following way:
Pokurskaya site (formation), number of layer(s) 1 or 19-20.
Fields 1-3 are oilfields currently in development and according to the Russian laws and company’s policy we can’t name them, because this is both government and company’s confidential information.
Point 7: ‘Stress-state model’ is much more than presented in the Figs. 5-7. Please correct titles and explain in the main text what is presented in each chart.
Response 7: Since all graphs give the same result – excess of effective (actual) stresses over rock strength, we think that explanations in the lines 274-277 are sufficient enough to prove the stated idea.
Point 8: Conclusion 1. – What do you mean by “permissible repressions”? – There is no sufficient discussion in the main text.
Response 8: Repression – excess of current pressure at the bottom of the well over reservoir pressure.
We believe this is our mistake to translate this term [repression] directly into English from Russian (Репрессия на пласт).
Can you please suggest which term is better for the explanation in the beginning of this Note?
Abnormal formation pressure or Overburden on formation?
We should also say that discussion of the drilling parameters is not the main purpose of the given article and they should be discussed in the different article.
Point 9: Conclusions 3 and 4. – There is no sufficient discussion in the main text.
Response 9: We deleted 3rd conclusion since this would take a lot of discussion and field data to be added and this is not the aim of the article.
We’ve added more field data to prove our point - regarding conclusion 4 (more data in lines 332-340).

Reviewer 2 Report
This paper presents a workflow for sand production analysis using combined numerical modeling, field data analysis and experiments. I have some a few comments as listed below.
- The English writing of this paper should be improved.
- Although the authors referred to some papers for the geomechanical modeling, please add a short description of how the modeling is done, what does Figure 4 show, and what does the software do in Section 2.1.
- It would be great to add a geological map to show where the three studied oil fields are and some other geological information.
- Please use “.” to represent decimal points instead of “,”.
- In the simulation study, it is just one case per field for the three fields. Some sensitivity studies should be done to account for the uncertainties in the parameters.
- In the experimental study, there also too few results and analysis. Only one figure is shown for the experiments. The authors should consider elaborating more on the experimental results.
Author Response
To whom it may concern,
We hope you and your family are healthy and safe during these uncertain and unprecedented times.
You can see all the changes that were made in the editor’s mode of MS Office Word.
Since there is more than one editor, you can see more changes than each single editor suggested or proposed in his/her review.
We will once again check the paper for mistakes and formatting issues;
Response to Reviewer 2 Comments
Point 1: The English writing of this paper should be improved.
Response 1: We have asked one of our colleagues with a high level of English to check the paper for mistakes. His corrections have been taken into account.
Point 2: Although the authors referred to some papers for the geomechanical modeling, please add a short description of how the modeling is done, what does Figure 4 show, and what does the software do in Section 2.1.
Response 2: Since this software is the property of the company and it is being developed by different departments and subsidiaries, we (as users) cannot explain in detail the algorithm behind the scenes, but we do have all the necessary rights to perform scientific studies with this software.
We believe that Figure 4 (in the current edit - Figure 5) is self-explanatory– it just states which data is used in the further modelling and how (with which techniques/equipment) this data is acquired.
We added a sentence regarding where this algorithm was taken from, this was a mistake we didn’t do it before.
Point 3: It would be great to add a geological map to show where the three studied oil fields are and some other geological information.
Response 3: According to the Russian laws and Company’s policy, we cannot name these oil fields or give any information about their location, we can, however add a regional map if you think that this is a necessary supplement to the paper.
You can also see that we didn’t name them in the Tables 1,2 and Figures 9, 10, 11.
Point 4: Please use “.” to represent decimal points instead of “,”.
Response 4: All “,” were corrected with “.” to express decimals.
Point 5: In the simulation study, it is just one case per field for the three fields. Some sensitivity studies should be done to account for the uncertainties in the parameters.
Response 5: This is a difficult task, because the graphs are circular and large, we can’t express this in the form of straight line to show the sensitivity graph.
The problem is that a calculation is made for each single angle (step = 1 degree). We can either make an average of the results, for example, in the ranges of 0-180 and 181-360 degrees, and lose some accuracy in the calculations and then build the sensitivity graph or build 360 sensitivity graphs for each angular position with 1 degree step or add more graphs with different bottom hole pressure values (so 6-9 more graphs, but then the paper will become much longer).
Point 6: In the experimental study, there also too few results and analysis. Only one figure is shown for the experiments. The authors should consider elaborating more on the experimental results.
Response 6: We prepared a different paper on the results of these laboratory experiments with much more data and analysis, where information/explanations regarding the process of sanding (dynamic) will be given with individual data on each of the experiments.
We should clarify that these results of prepack-tests are supplementary to the modelling that confirm our thesis of rock destruction during well exploitation with different pressure gradients.
